# Spatial Analysis of HIV Determinants Among Females Aged 15–34 in KwaZulu Natal, South Africa: A Bayesian Spatial Logistic Regression Model

**DOI:** 10.3390/ijerph22030446

**Published:** 2025-03-17

**Authors:** Exaverio Chireshe, Retius Chifurira, Knowledge Chinhamu, Jesca Mercy Batidzirai, Ayesha B. M. Kharsany

**Affiliations:** 1School of Mathematics, Statistics and Computer Science, College of Agriculture, Engineering and Science, University of KwaZulu-Natal, Durban 4001, South Africa; 2Centre for the AIDS Programme of Research in South Africa (CAPRISA), Doris-Duke Medical Research Institute, Nelson R Mandela School of Medicine, University of KwaZulu-Natal, Durban 4001, South Africa; ayesha.kharsany@caprisa.org

**Keywords:** HIV prevalence, Bayesian logistic regression, Kulldorf’s spatial scan statistics, odds ratios, spatial clustering

## Abstract

HIV remains a major public health challenge in sub-Saharan Africa, with South Africa bearing the highest burden. This study confirms that KwaZulu-Natal (KZN) is a hotspot, with a high HIV prevalence of 47.4% (95% CI: 45.7–49.1) among females aged 15–34. We investigated the spatial distribution and key socio-demographic, behavioural, and economic factors associated with HIV prevalence in this group using a Bayesian spatial logistic regression model. Secondary data from 3324 females in the HIV Incidence Provincial Surveillance System (HIPSS) (2014–2015) in uMgungundlovu District, KZN, were analysed. Bayesian spatial models fitted using the Integrated Nested Laplace Approximation (INLA) identified key predictors and spatial clusters of HIV prevalence. The results showed that age, education, marital status, income, alcohol use, condom use, and number of sexual partners significantly influenced HIV prevalence. Older age groups (20–34 years), alcohol use, multiple partners, and STI/TB diagnosis increased HIV risk, while tertiary education and condom use were protective. Two HIV hotspots were identified, with one near Greater Edendale being statistically significant. The findings highlight the need for targeted, context-specific interventions to reduce HIV transmission among young females in KZN.

## 1. Background

Human immunodeficiency virus (HIV) remains a major public health concern in South Africa, with an estimated 7.8 million people living with HIV and a 19.85% prevalence rate among adults aged 15–49 as of 2021 [1]. KwaZulu-Natal (KZN) is the most affected province, recording the highest HIV prevalence and facing severe socio-economic impacts [2].

Women aged 15–34 are particularly vulnerable, accounting for a substantial proportion of new infections due to biological susceptibility, societal pressures, and economic hardships [3,4]. Studies indicate that inconsistent condom use, early sexual debut, substance abuse, and concurrent sexually transmitted infections (STIs) further increase their risk of HIV acquisition [5,6]. Additionally, socio-economic determinants such as poverty, unemployment, transactional sex, and intergenerational relationships heighten their vulnerability [7]. Intimate partner violence (IPV) further exacerbates this risk by restricting women’s ability to negotiate safer sexual practices [8].

Geospatial disparities in HIV prevalence across KZN, particularly in peri-urban and rural areas, highlight challenges such as limited healthcare access and high poverty rates [9]. Studies using spatial epidemiology have identified clusters of high HIV prevalence, underscoring the need for geographically targeted interventions that address structural determinants of HIV risk [10]. However, existing research does not adequately account for spatial dependencies and geographic heterogeneity, limiting the ability to develop spatially adaptive public health strategies.

To address these gaps, this study employs Bayesian spatial logistic regression, a robust framework for analysing spatial, demographic, and individual-level factors influencing HIV prevalence. Unlike traditional logistic regression, this approach incorporates spatial dependencies and heterogeneity, which are critical for understanding the geographic variability of HIV burden across KZN [11]. Additionally, Bayesian methods integrate prior knowledge with observed data, yielding more stable estimates, particularly in regions with sparse data. The effectiveness of Bayesian spatial models has been proven in various epidemiological contexts. For example, Bayesian semi-parametric regression has been applied to study HIV prevalence among men in Kenya, incorporating structured and unstructured spatial effects to enhance model accuracy [12]. Similarly, Bayesian spatial modelling has been used to examine tuberculosis–HIV co-infection in Ethiopia, revealing significant geographical heterogeneity in disease distribution [13]. Other studies in sub-Saharan Africa have leveraged Bayesian hierarchical models to assess the spatial distribution of HIV risk factors, emphasising the role of socio-economic and behavioural determinants in shaping HIV prevalence patterns.

An extension of these methods, structured additive models, enhances Bayesian spatial approaches by incorporating non-linear effects of continuous variables alongside spatial random effects [14,15]. This flexibility allows for a more nuanced analysis, integrating both individual and area-level risk factors to produce accurate spatial estimates and better identify high-risk clusters [16]. Such models are particularly valuable in regions like KwaZulu-Natal, where complex interactions between socio-demographic and geographic factors contribute to HIV transmission.

Spatial models such as the Besag–York–Mollié (BYM) model have been widely used in HIV research to analyse structured and unstructured spatial effects, often implemented efficiently using Integrated Nested Laplace Approximation (INLA) [17,18]. These models have helped uncover geographic disparities in HIV prevalence, facilitating the identification of high-risk clusters and informing public health interventions [16].

This study aimed to investigate the spatial distribution and key demographic, behavioural, and socio-economic factors associated with HIV prevalence among female youth in KwaZulu-Natal using a Bayesian spatial logistic regression framework with a structured additive model. Despite the increasing use of Bayesian spatial models in HIV research, a few studies have simultaneously accounted for both micro-level (individual risk factors) and macro-level (spatial dependencies) determinants of HIV prevalence, specifically among female youth in KwaZulu-Natal. By integrating these elements, our study provides a more comprehensive understanding of HIV transmission dynamics, offering insights that can support targeted public health interventions in high-burden areas.

## 2. Methodology

### 2.1. Study Area Location

Figure 1A,B below depict the location of the study area within uMgungundlovu District and the location of the two sub-districts of KwaZulu-Natal Province, namely, Vulindlela (western part) and the Greater Edendale (eastern part), respectively.

### 2.2. Sources of Data and Study Population

This study conducted a secondary analysis of data from the HIV Incidence Provincial Surveillance System (HIPSS), a population-based surveillance study carried out between 11 June 2014 and 18 July 2015 in the rural Vulindlela and peri-urban Greater Edendale areas of uMgungundlovu District, KwaZulu-Natal, South Africa. The HIPSS dataset is widely recognised for its robust design and methodological rigour in estimating HIV incidence and prevalence.

To ensure representativeness, HIPSS employed a multi-stage probability sampling strategy. From a total of 600 enumeration areas (EAs), 591 EAs with over 50 households were included. Of these, 221 EAs were randomly selected. Within each EA, households were systematically sampled at random, ensuring an unbiased selection process. Only one age-eligible individual per household was randomly chosen for participation, following written informed consent. Geographic coordinates of each randomly selected household were recorded using Global Positioning Systems (GPS) to ensure spatial accuracy and avoid selection bias.

To ensure data accuracy and integrity, HIPSS implemented rigorous quality control measures throughout the study. Data quality was monitored daily for the first month, then monthly for six months, and subsequently at three-month intervals. The Mobenzi Researcher system (Durban, South Africa) enabled real-time tracking of field teams to ensure adherence to protocols and accurate data collection. Automated quality checks allowed for immediate anomaly detection and corrective actions. Additionally, laboratory results from peripheral blood samples for HIV testing were integrated into the dataset, ensuring comprehensive epidemiological data. All real-time data were centrally managed following stringent quality checks, minimising errors and ensuring completeness. By employing systematic probability sampling, real-time tracking, and multi-level quality control, the HIPSS dataset provided highly reliable and representative data on females aged 15–34 in KwaZulu-Natal. This ensured that our study’s findings accurately reflect the demographic, behavioural, and socio-economic factors influencing HIV prevalence in this population.

Youth is often defined as individuals aged 15–24 (United Nations). However, this study adopts a broader definition, encompassing individuals aged 15–34, in alignment with regional demographic trends and South African policy frameworks [19]. This expanded age range captures critical life transitions influencing HIV risk, including adolescence, early adulthood, and early middle age. In South Africa, young adults up to 34 face significant socio-economic and health vulnerabilities due to factors such as unemployment, prolonged education, and delayed family formation. Given these realities, defining youth as 15–34 ensures that both adolescent and young adult populations are adequately represented, reflecting the epidemiological significance of HIV risk in KwaZulu-Natal.

A total of 9812 participants aged 15–49 years were enrolled in the HIPSS survey (6265 females and 3547 males). Among the 6265 females, 4144 were aged 15–34 years, and of these, 3324 were included in our study after excluding participants with incomplete HIV status or missing key demographic variables. Missing data were addressed using a complete-case analysis approach, where cases with missing values were removed. While this method ensures that only participants with complete data contribute to the analysis, it may introduce selection bias if the excluded cases differ systematically from those included. However, given the relatively low proportion of missing data, the impact on the overall findings is expected to be minimal.

### 2.3. Study Variables

The dependent variable in this study was “HIV prevalence”, which was defined as the ratio of the number of HIV-positive participants in an enumeration area to the total number of participants in the same enumeration area. In our analysis, we used unweighted HIV prevalence since we were focused on detecting geographic locations where spatial clustering of HIV prevalence occurs.

HIV status among participants in the study population was categorised as a binary outcome:(1)$$y_{ij}=\left\{\begin{array}{c} 1\;\;\;for\;a\;participant\;who\;tested\;positive\\ \;0\;\;\;for\;a\;participant\;who\;tested\;negative \end{array}\right.$$

The socio-demographic, behavioural, and biological covariates included in the model were selected based on their epidemiological relevance, data availability, and statistical significance. Variables known to influence HIV risk, as identified in previous studies and public health reports, were prioritised to ensure the model captured key determinants. Only variables with minimal missing data were included to maintain robustness and avoid biases arising from incomplete information. To ensure proper adjustment for confounding, univariate analyses were first conducted to assess the association of each variable with HIV prevalence. Only significant predictors were retained for the multivariate model. The selected covariates included age, level of education, marital status, main income, alcohol consumption, history of tuberculosis (TB) or sexually transmitted infections (STIs), number of sexual partners, condom use, forced first sex, pregnancy history, financial instability (running out of money and meal cuts), mobility (being away from home and duration in the community), and access to healthcare. To control for multicollinearity, the Variance Inflation Factor (VIF) was calculated for all covariates, with all values remaining below 1.5, indicating minimal collinearity. A stepwise selection approach was applied during model fitting to exclude non-significant or redundant covariates, ensuring a more parsimonious and interpretable model. Additionally, spatial dependency was explicitly modelled to account for unmeasured geographic factors, reducing the risk of omitted variable bias and improving the accuracy of the findings.

### 2.4. Spatial Autocorrelation

The Bayesian spatial logistic regression model was chosen for this study due to its ability to account for spatial dependencies, heterogeneity, and uncertainty in HIV prevalence among females aged 15–34 in KwaZulu-Natal. Unlike traditional frequentist models that assume independence between observations, the Bayesian framework explicitly incorporates spatial autocorrelation, improving the accuracy of geographic risk estimation. Given the presence of spatial clustering in HIV prevalence across the study region, this approach is particularly relevant.

To assess spatial patterns and validate model assumptions, we conducted a spatial analysis using enumeration areas (EAs) with geo-referenced boundaries as the spatial units linked to HIV prevalence data. Since neighbouring observations tend to exhibit similar values, spatial models help distinguish between systematic geographic patterns and random spatial variation [20]. We quantified spatial autocorrelation using Moran’s I statistic and Geary’s C statistic, confirming the presence of spatial structure in the data.

#### 2.4.1. Global Moran’s Index Statistic

The Global Moran’s Index measures overall spatial autocorrelation across a study area, indicating the presence, strength, and direction of spatial patterns. Positive autocorrelation occurs when neighbouring enumeration areas have similar values, while negative autocorrelation suggests contrasting values. When spatial patterns are random, the index approaches zero [21,22,23,24].

The Moran’s Index is calculated as follows:(2)$$I=\frac{n}{\Theta }\times \frac{\sum_{i=1}^{n}\sum_{j=1}^{n}\Theta_{ij}\left(y_{i}-\bar{y}\right)\left(y_{j}-\bar{y}\right)}{\sum_{i=1}^{n}{\left(y_{i}-\bar{y}\right)}^{2}}$$
where n is the total number of enumeration areas, $y_{i}$ is the value of the variable at location i, $\bar{y}$ is the mean of the variable y across all enumeration areas, $\Theta_{ij\;}$ is the spatial weight between enumeration area i and enumeration area j, and Θ indicates the sum of all spatial weights.

#### 2.4.2. Geary’s C Statistic

Geary’s C evaluates spatial autocorrelation by assessing similarity or dissimilarity between values at neighbouring locations. Unlike Moran’s Index, it is sensitive to local variations [25,26]. The formula for Geary’s C is as follows:(3)$$C=\frac{\left(N-1\right)\sum_{i=1}^{N}\sum_{j=1}^{N}\Theta_{ij}{\left(y_{i}-y_{j}\right)}^{2}}{2\Theta \sum_{i=1}^{N}{\left(y_{i}-\bar{y}\right)}^{2}}$$
where *N* is the total number of enumeration areas (locations), $y_{i}$ and $y_{j}$ are the values of the variable of interest at locations *i* and *j*, $\bar{y}$ is the mean of the variable across all locations, $\Theta_{ij}$ is the spatial weight between location *i* and location *j*, and Θ is the sum of all $\Theta_{ij}$. Values of *C* < 1 indicate positive spatial autocorrelation, *C* > 1 indicate negative spatial autocorrelation, and *C* = 1 implies no autocorrelation [27].

To further identify and interpret spatial clusters, we employed Kulldorff’s spatial scan statistic (SaTScan). Unlike Moran’s I and Geary’s C, which assess overall spatial patterns, SaTScan detects significant high-risk (hot-spots) and low-risk (cold-spots) clusters by scanning circular windows across the study area. This approach allows for the precise localisation of areas with significantly higher or lower HIV prevalence, thereby informing targeted public health interventions [28]. In this study, SaTScan was used to identify clusters of HIV prevalence among young women aged 15–34. The decision to analyse this broader age group, rather than focusing solely on adolescents aged 15–19, was driven by the need to capture overall spatial trends in HIV distribution and account for the cumulative nature of HIV infection. Since HIV is a chronic condition, prevalence increases with age due to both new infections and the accumulation of cases over time. Analysing the entire 15–34 age group provides a more comprehensive view of geographic variations in risk and potential structural drivers of HIV transmission.

### 2.5. Bayesian Logistic Regression Models

Bayesian logistic regression is a powerful method for modelling binary outcomes, such as disease presence, by estimating posterior distributions of regression parameters. This approach combines prior beliefs with observed data, resulting in posterior distributions that reflect both sources of information [17,29]. In this study, the Bayesian spatial logistic regression model accounts for both individual-level factors (e.g., age, education, and behavioural risk factors) and spatial dependencies through a structured random-effects component. The spatial effect is modelled using a structured spatial component, which captures spatial autocorrelation by borrowing strength from neighbouring areas. This approach ensures that unobserved neighbourhood-level influences on HIV prevalence, such as healthcare access, socio-economic disparities, and localised prevention efforts, are accounted for beyond the effects of individual-level risk factors alone. The persistence of spatial clustering, even after adjusting for individual factors, suggests that geographic factors contribute independently to HIV risk.

The binary outcome $Y_{i}\in \{0;1\}$ follows a Bernoulli distribution:

$Y_{i\;}\sim$ Bernoulli ($p_{i}$), where $p_{i}$ is the probability that $Y_{i}=1$, linked to the linear predictor $\varphi_{i}$ by the logistic function:(4)$$p_{i}=\frac{exp\;(\varphi_{i})}{1+exp\;(\varphi_{i})}$$
and(5)$$\varphi_{i}=\beta_{0}+X_{i}^{T}\beta$$
where $\beta_{0}$ is the intercept, $X_{i}$ is the vector of covariates for observation i, and β is the vector of regression coefficients. The likelihood function of *N* observations is expressed as follows:(6)$$L\left(\beta \right)=\prod_{i=1}^{N}p_{i}^{Y_{i}}{\left(1-p_{i}\right)}^{1-Y_{i}}$$

Bayesian spatial logistic regression extends this framework by incorporating spatial dependencies, enabling the analysis of structured and unstructured spatial variability in binary data such as disease prevalence [30].

The model is given by $Y_{i\;}\sim$ Bernoulli ($p_{i}$), with the probability $\;p_{i}$ linked to the linear predictor $\varphi_{i}$:(7)$$p_{i}=\frac{exp\;(\varphi_{i})}{1+exp\;(\varphi_{i})}$$

The linear predictor includes spatial random effects:(8)$$\varphi_{i}=\beta_{0}+X_{i}^{T}\beta +\theta_{i}$$
where $\theta_{i}$ represents the spatial random effects.

Spatial dependencies are captured using priors like the conditional autoregressive (CAR) model, intrinsic CAR (ICAR) model, or Gaussian process (GP) model, allowing robust modelling of spatially correlated binary outcomes [31,32].

### 2.6. Prior Distributions

In Bayesian analysis, prior distributions represent beliefs about parameters before observing data and are combined with likelihood functions to obtain posterior distributions [17,32,33]. Priors are essential in hierarchical spatial models, especially with small sample sizes or variable data, and help regularise the model [34].

Choosing priors involves balancing prior knowledge and non-informativeness. Informative priors guide inference when prior knowledge is available, while weakly informative or non-informative priors are used when prior knowledge is absent. In this study, non-informative priors were used for regression coefficients and random-effects variances due to a lack of prior knowledge.

Penalised complexity (PC) priors were applied to the precision parameter of the random effects. These priors balance model simplicity and complexity, avoiding issues like overfitting and computational problems associated with flat priors [35,36]. The PC prior for precision ρ is expressed as follows:(9)$$\pi \left(\rho \right)=\nu e^{-\nu \sqrt{\rho }}$$
with(10)$$\nu =\frac{log\left(\alpha \right)}{U}$$
and(11)$$\rho =\frac{1}{\sigma^{2}}$$
where ρ is the precision, U is the upper bound for the standard deviation σ of the random effect, and α is the probability that σ>U.

### 2.7. Posterior Distributions and Point Estimates

The posterior distribution contains complete information about parameter estimates, summarised using point estimates and credible intervals. Point estimates include the posterior mean, posterior mode, and posterior median, which are used for inference and prediction.

The posterior mean is the expected value of the parameter under the posterior distribution. It is a common estimate, especially when the posterior is symmetric. For a parameter β, it is given by:(12)$${\hat{\beta }}_{mean}=E\left[\beta |y\right]=\int \beta P\left(\beta |y\right)d\beta$$

The posterior mode, also known as the maximum a posteriori (MAP) estimate, is the mode of the posterior distribution, i.e., the value of β that maximises $P\left(\beta |y\right)$ and is expressed as:(13)$${\hat{\beta }}_{MAP}={arg\;max}_{\beta }P\left(\beta |y\right)$$
where ${arg\;max}_{\beta }$ indicates finding the value of β that maximises this posterior probability. The MAP estimate is often used when the posterior is skewed, but it can be sensitive to the choice of the prior.

The posterior median is a robust point estimate that divides the posterior distribution into two equal parts. It is less sensitive to outliers compared to the mean or mode.

Credible intervals provide the range where the parameter likely falls with a given probability. A 95% credible interval means there is a 95% probability that the true parameter lies within the interval:(14)$$P\left(\beta_{lower}<\beta <\beta_{upper}|y\right)=0.95$$

Unlike frequentist confidence intervals, credible intervals offer a direct probabilistic interpretation.

### 2.8. Bayesian Spatial Logistic Regression Models Applied

Bayesian logistic regression incorporates prior beliefs and spatial dependencies. Below are the applied models.

#### 2.8.1. Unstructured Bayesian Spatial Logistic Regression Model

This model accounts for heterogeneity by incorporating independent and identically distributed random effects, assuming no spatial dependency [17,37]. It is defined as $Y_{i\;}\sim$ Bernoulli ($p_{i}$), with(15)$$logit\left(p_{i}\right)=\beta_{0}+X_{i}^{T}\beta +u_{i}$$
where $u_{i}$ denotes the unstructured random effects and $u_{i}\sim Ν\left(0,{\sigma^{2}}_{u}\right)$.

#### 2.8.2. Structured Bayesian Spatial Logistic Regression Model

This model incorporates spatial dependence using a structured random field, improving predictions by considering the influence of nearby locations. It is defined as follows:(16)$$Y_{i\;}\sim \mathrm{Bernoulli}\;(p_{i}),\;\mathrm{with}\;logit\left(p_{i}\right)=\beta_{0}+X_{i}^{T}\beta +\theta_{i}$$
where $\theta \sim CAR\left(W,\;Τ\right)$ and the conditional autoregressive (CAR) model for θ assumes$$\theta_{i}|\theta_{-i},Τ\sim Ν\left(\frac{1}{\eta_{i}}\sum_{j\in neigh\left(i\right)}\theta_{j},\frac{1}{{Τ\eta }_{i}}\right),$$
where $\theta_{i}$ indicates the spatially structured random effect at location i, $\eta_{i}$ represents the number of neighbours of location i, W is a spatial adjacency matrix, and Τ is the precision parameter [14,31,38].

### 2.9. Model Selection Criteria

To ensure the robustness of the Bayesian spatial logistic regression model, we conducted thorough validation and sensitivity analyses. Model selection was based on the following model selection criteria: deviance information criteria (DIC) [39], the effective number of parameters (pD), the mean deviance ($\tilde{D}$), and the Watanable–Akaike information criteria (WAIC) [40]. Lower DIC, $\tilde{D}$, and WAIC values and a higher pD value suggest a better model fit. Hence, the best-fitting model was selected based on the smallest DIC, $\tilde{D}$, and WAIC and the highest pD.

### 2.10. Model Diagnostics

After selecting the best-fitting model, we assessed its adequacy using residual plots and normal Q–Q plots. A well-fitted model should have residuals symmetrically distributed around zero, with no clear pattern or trend and constant variance [41,42,43]. Deviations from normality in the Q–Q plot suggest that residuals do not follow a normal distribution. We also examined spatial autocorrelation in residuals using Moran’s I statistic, Geary’s C statistic, and the variogram plot to verify whether the spatial structure was adequately captured. High spatial autocorrelation in residuals indicates the model failed to fully account for spatial dependencies [44,45]. Significant Moran’s I and Geary’s C values suggest poor model fit. Increasing semi-variance with distance indicates spatial autocorrelation, suggesting an inadequate model. Flat variogram suggests spatially uncorrelated residuals, indicating a well-fitted model [44,46]. Additionally, posterior density plots were examined for model validity, reliability, and stability. A smooth, unimodal density plot indicates a well-fitting model, while a multimodal plot may suggest model ambiguity or data issues [17].

### 2.11. Software and Implementation

The Bayesian spatial logistic regression models were implemented using the Integrated Nested Laplace Approximation (INLA) method [14,47] in R (version 4.4.0). The following R packages were used: “INLA”, “sf”, “sp”, “spdep”, and “dplyr” packages. Spatial relationships between enumeration areas were established using a spatial weight matrix, with neighbours identified via Queen’s contiguity. Additionally, Kulldorff’s spatial scan statistics were applied using SaTScan (version 10.1.3).

## 3. Empirical Results

Summary statistics for the HIV prevalence rates for all the covariates included in the study are depicted in Table 1. While the summary statistics provide an initial indication of associations, the Bayesian model results are prioritised due to their robustness in adjusting for spatial correlations and confounding effects. This approach ensures that our conclusions are based on a more comprehensive analysis of the data.

There were 3324 females who were included in this research and 1576 individuals were HIV positive, giving us an overall HIV prevalence of 47.4% (95% CI: 45.7–49.1) (*p*-value < 0.0001). We noticed that HIV prevalence increased as age increased, and it was 20.4% (95% CI: 16.8–24.5), 37% (95% CI: 34.2–40.0), 54% (95% CI: 50.8–57.1), and 67.5% (95% CI: 64.2–70.8) for age groups 15–19, 20–24, 25–29, and 30–34, respectively (*p*-value < 0.0001). Considering education level, individuals with primary education had the highest HIV prevalence of 70.6% (95% CI: 59.7–80.0), followed by those with no schooling with 55.6% (95% CI: 44.1–66.6) (*p*-value < 0.0001). Participants who had no source of income had the highest HIV prevalence of 50.5% (95% CI: 43.4–57.6) (*p*-value = 0.169).

Table 1 represents the HIV prevalence rates for all the covariates included in the study.

Looking at the marital status covariate, participants who were divorced and those who were separated but still legally married had the highest HIV prevalence of 100% (95% CI: 15.8–100.0), followed by participants who were single but had been living with someone as a husband/wife before with an HIV prevalence of 63.7% (95% CI: 55.5–71.8). The *p*-value for the marital status covariate is 0.000181. The HIV prevalence was higher among participants who were once diagnosed with TB, 63% (95% CI: 54.6–70.8) compared to those who were not diagnosed with TB, 46.7% (95% CI: 45.5–48.5) (*p*-value = 0.000365). Participants who indicated that they were not using condoms as a prevention method had a higher HIV prevalence, 58.1% (95% CI: 47.0–68.7), compared to those who were using condoms, 47.1% (95% CI: 45.4–48.9) (*p*-value = 0.056). Classified by the number of sexual partners, HIV prevalence increased as the number of partners increased, and it was 45.5% (95% CI: 43.6–47.3) for participants with one partner, 51.6% (95% CI: 45.9–57.3) for participants with two partners, and 67.5% (95% CI: 60.6–73.8) for participants with three partners (*p*-value < 0.0001). HIV prevalence for participants who did not consume alcohol was slightly lower, 45.8% (95% CI: 43.9–47.6), compared to those who were consuming alcohol, 58.5% (95% CI: 53.7–63.3) (*p*-value < 0.0001). Participants who were diagnosed with STIs had a higher HIV prevalence of 63% (95% CI: 56.2–69.4), compared to 46.3% (95% CI: 44.5–48.1) for participants who were not diagnosed with STIs (*p*-value < 0.0001). Based on the forced first sex covariate, participants who had forced first sex had the highest HIV prevalence of 54.7% (95% CI: 43.5–65.4) (*p*-value = 0.246837). Participants who were away from home had a higher HIV prevalence of 50.1% (95% CI: 44.8–55.5), compared to those who were not away from home (*p*-value = 0.407053). For the length in community covariate, the highest HIV prevalence of 60% (95% CI: 14.7–94.7) was recorded for participants who did not respond, and the lowest HIV prevalence of 46.7% (95% CI: 44.8–48.7) was observed for those participants who were always in the community (*p*-value = 0.448). The HIV prevalence was higher among participants who accessed health care, 50.5% (95% CI: 47.7–53.4) compared to those who did not respond, 33.3% (95% CI: 4.3–77.7) and those who did not access health care, 45.6% (95% CI: 43.4–47.8) (*p*-value = 0.018). Considering the run out of money covariate, participants who ran out of money had the highest HIV prevalence of 49.1% (95% CI: 45.2–52.9) (*p*-value = 0.618). The HIV prevalence was slightly higher among participants who had meal cuts, 47.7% (95% CI: 43.7–51.8), compared to those who had no meal cuts, 47.5% (95% CI: 45.6–49.4) (*p*-value = 0.516). Lastly, participants who once became pregnant had a higher HIV prevalence of 50.4% (95% CI: 48.4–52.3) compared to those who had never become pregnant, 37.4% (95% CI: 33.9–40.9) (*p*-value < 0.0001).

The HIV prevalence also varied among enumeration areas (ranging between 0 and 100%). The geographical distribution of HIV prevalence by enumeration areas is shown in Figure 2. This map was created using ArcGIS Pro software (version 3.4) with the application of the “tidyverse”, “sf”, and “tmap” packages in R software (version 4.4.0).

The result for Moran’s Index statistic of HIV prevalence was 0.707 with a *p*-value < 0.001, indicating a very strong positive spatial autocorrelation in the wards of uMgungundlovu District (Table 2). The positive and statistically significant Moran’s Index value supports that there are clusters of high and low HIV prevalence areas within the study region, suggesting a non-random spatial pattern. The positive Moran’s Index also suggests that the HIV prevalence in any two spatial neighbouring wards tended to have similar HIV prevalence.

Furthermore, the findings from Geary’s C test statistic support the results from Moran’s Index statistic as they both reveal consistent evidence of spatial heterogeneity in HIV prevalence within uMgungundlovu District. The summary statistics results for Moran’s Index statistic and Geary’s C statistic are displayed in Table 2 below.

As shown in Table 2, both Moran’s I and Geary’s C indicate significant and strong positive spatial autocorrelation in HIV prevalence. These results confirm spatial heterogeneity, suggesting that HIV prevalence is not randomly distributed but influenced by underlying spatial processes or risk factors in uMgungundlovu District. However, while Moran’s I and Geary’s C detect spatial autocorrelation, they do not differentiate hotspots from cold spots. To address this, Kulldorff’s spatial scan statistics were applied, identifying two clusters of HIV prevalence. The spatial distribution of these clusters is visualised in Figure 3.

Cluster 1, a hotspot with a 2.53 km radius, had an HIV prevalence of 48.4%, a relative risk (RR) of 1.22, and a *p*-value of 0.025, indicating a 22% higher risk of HIV infection within the cluster compared to areas outside it. The low *p*-value (<0.05) confirms that this increased risk is statistically significant, meaning it is unlikely to be due to random chance. This cluster was located around Greater Edendale.

Cluster 2, another hotspot with a 2.28 km radius, had an HIV prevalence of 49.6% and an RR of 1.28, suggesting a 28% higher HIV risk within the cluster. However, its *p*-value was 0.467, which is above the conventional 0.05 threshold for significance. This means the observed higher risk in this cluster could be due to random variation rather than a true spatial effect. This cluster covered Nadi, KwaMbanjwa, Zayeka, KwaMtogotho, KwaNxamalala, and Henley.

To identify factors associated with HIV prevalence, Bayesian spatial logistic regression was applied, considering socio-demographic, behavioural, and biological factors. Most covariates were statistically significant at the 5% level for both models. Model selection was based on DIC, pD, $\tilde{D}$, and WAIC, as shown in Table 3.

Based on WAIC, DIC, and pD, the structured model emerged as the best model. It has lower DIC, pD, and WAIC values, as shown in Table 3, compared to the unstructured model. The structured model strikes the best balance between model fit, complexity, and predictive accuracy, making it the optimal choice. Hence, the results of this research are based on the structured model.

Adjusted odds ratios (ORs), together with their corresponding 95% credible intervals (CI) for the participants’ characteristics, are displayed in Table 4. These values were obtained from the fitted structured Bayesian spatial logistic regression model implemented in INLA.

Most of the covariates included in the study were significant, providing insights into the factors associated with HIV prevalence. Covariate levels with 95% credible intervals, including 1, were not statistically significant, and, as a result, we did not consider them as predictors of HIV prevalence in our study.

The findings revealed that the odds of HIV prevalence for participants in the age groups 20–24, 25–29, and 30–34 were 2.337 (OR = 2.337, 95% CI: 1.791–3.053), 4.745 (OR = 4.745, 95% CI: 3.611–6.234), and 9.198 (OR = 9.198, 95% CI: 2.883–12.293) times higher than that of age group 15–19, respectively.

Considering education, participants with incomplete secondary were 1.405 (OR = 1.405, 95% CI: 1.195–1.652) times more likely to be HIV-infected compared to those with complete secondary. Participants with no schooling were 1.718 (OR = 1.718, 95% CI: 1.065–2.773) times more likely to be HIV-infected compared to participants with complete secondary. Also, participants with primary education were 2.612 (OR = 2.612, 95% CI: 1.597–4.276) times more likely to be HIV-infected compared to participants with complete secondary. Importantly, participants with tertiary education were 0.534 (OR = 0.534, 95% CI: 0.391–0.728) times less likely to be HIV-infected compared to those with complete secondary.

Results based on the main income covariate revealed that participants with salary and or wage had a reduced risk of getting infected with HIV (OR = 0.706, 95% CI: 0.522–0.956), compared to those with no source of income.

We found that individuals who were legally married had a reduced risk of getting infected with HIV (OR = 0.371, 95% CI: 0.150–0.919), compared to those who were divorced. The results also revealed that there was a higher likelihood of being infected by HIV among individuals who were diagnosed with TB (OR = 1.799, 95% CI: 1.247–2.594), compared to those who never suffered from TB. We also discovered that there was a higher likelihood of getting HIV infection among participants who were diagnosed with STIs (OR = 1.694, 95% CI: 1.245–2.303), compared to those who were not diagnosed with STIs.

Considering the number of sexual partners, there was a higher likelihood of being HIV-infected among participants who had three or more sexual partners (OR = 1.765, 95% CI: 1.275–2.445), compared to those who had one partner. Results based on alcohol consumption showed that individuals who consumed alcohol had odds of HIV prevalence that was 1.644 (OR = 1.644, 95% CI: 1.310–2.168) times higher than those who were not consuming alcohol. Lastly, we found that using condoms as a prevention method reduced the risk of being HIV-infected (OR = 0.552, 95% CI: 0.348–0.874) compared to not using condoms.

The results above indicate that age group, education levels, the source of income, and marital status, along with behaviours like alcohol use, condom use, and having multiple sexual partners, are the key predictors of HIV prevalence. Also, being diagnosed with sexually transmitted infections (STIs) and TB increases the chances of getting infected with HIV.

After fitting the model, the smoothed HIV prevalence rates were calculated and are displayed in Figure 4.

Comparing the HIV prevalence intervals in Figure 2 (unsmoothed prevalence rates) and Figure 4 (smoothed prevalence rates), we observe that the intervals differ. However, areas with high HIV prevalence in the unsmoothed data remain high-prevalence areas in the smoothed data, indicating consistency in spatial patterns.

Model performance was assessed using a residuals plot and normal Q–Q plot for model adequacy and Moran’s I, Geary’s C statistic, and the variogram plot to evaluate spatial autocorrelation in residuals. Figure 5 below displays the residuals plot.

Figure 5 displays residuals that are symmetrically distributed around zero, showing no clear pattern, and having constant variance. The plot suggests that there is no systematic bias in the model’s predictions, implying that the model fits the data well. Figure 6 below displays the Q-Q plot of the residuals.

The Q–Q plot in Figure 6 shows an S-shaped pattern, indicating deviation from normality with heavier tails. However, in spatial modelling, residuals are not always expected to be normally distributed due to inherent spatial dependencies. This characteristic is well documented in spatial statistics [32,38,44,48,49].

The global Moran’s I statistic for residuals was 0.0009971 (*p* = 0.4549), suggesting no significant spatial autocorrelation. This indicates that the structured model has adequately captured the spatial structure in the data.

Similarly, Geary’s C statistic was 1.0010397 (*p* = 0.5349), further confirming that residuals are not spatially autocorrelated. Since both Moran’s I and Geary’s C suggest no significant spatial dependence, the model appears to fit well.

Additionally, the variogram plot in Figure 7 provides strong evidence that residuals are spatially uncorrelated, further supporting the model’s adequacy.

The plot shows a flat semi-variance around 0.20, indicating no spatial autocorrelation in the residuals. This suggests that the residuals are spatially independent. If spatial dependence were present, the semi-variance would increase with distance, which was not observed.

The posterior density plots for statistically significant regression parameters in Figure 8 display smooth curves with single peaks, indicating stability and proper model convergence.

Based on the spatial autocorrelation tests applied, the results revealed that the residuals were not spatially autocorrelated, implying that the structured model was appropriate and had captured the spatial structure in our data. This is also supported by smooth and unimodal plots displayed by the posterior density plots.

## 4. Discussion

This study employed a Bayesian spatial logistic regression approach to examine the prevalence and risk factors associated with HIV/AIDS among female youth in KwaZulu-Natal, South Africa. The findings indicate significant spatial clustering of HIV prevalence, with socio-demographic, behavioural, and health-related factors playing a crucial role in infection risk.

Age emerged as a key determinant of HIV prevalence. Participants aged 20–24, 25–29, and 30–34 faced significantly higher odds of HIV infection compared to those aged 15–19. This reflects the disproportionate burden of HIV among young adult women, often driven by power imbalances in relationships, transactional sex, and limited access to preventive services [9,50,51].

As expected, HIV prevalence increased with age, which is consistent with the chronic nature of the infection and cumulative exposure over time. The lower prevalence observed in the 15–19 age group may suggest reduced new infections, potentially due to recent prevention efforts. However, given that this is a cross-sectional study, we cannot directly assess incidence trends. Comparing the prevalence in this age group with previous survey rounds would provide better insight into whether new infections are indeed decreasing.

Education also played a critical role in HIV risk. Lower educational attainment was strongly associated with higher HIV prevalence, whereas tertiary education was protective. These findings underscore the importance of education in empowering young women with knowledge about HIV prevention and increasing their ability to make informed decisions about their sexual health [52,53,54]. Socio-economic factors, particularly income source, were significant. Female youth earning a salary or wage were less likely to be HIV positive, highlighting the protective role of financial independence and economic empowerment [55,56].

Risky sexual behaviours, including multiple sexual partners and inconsistent condom use, were significant predictors of HIV infection. These findings align with studies showing that such behaviours amplify the risk of HIV transmission in high-prevalence settings [57,58]. Alcohol use was also associated with higher odds of HIV infection, consistent with evidence that alcohol consumption impairs judgement and increases engagement in risky sexual behaviours [59,60,61]. Given the strong link between alcohol use and HIV risk, intervention strategies should incorporate behavioural change programmes focusing on alcohol reduction and safer sexual practices.

Co-infections with tuberculosis (TB) and sexually transmitted infections (STIs) were strongly associated with HIV infection. These co-morbidities exacerbate vulnerability to HIV, emphasising the need for integrated healthcare approaches that address HIV and other infectious diseases simultaneously. Strengthening TB and STI screening programmes within HIV care services is crucial for improving health outcomes [62,63,64].

Education, financial stability, and consistent condom use were identified as protective factors. Higher education levels provided young women with better awareness of HIV risks and preventive measures, while financial stability reduced dependence on transactional sex, which is a known risk factor for HIV acquisition [56,65]. Legal marriage was also found to be protective, potentially due to increased relationship stability and reduced exposure to high-risk sexual networks [66,67].

The structured additive model revealed significant spatial variations in HIV prevalence, emphasising the role of geographic location in HIV risk among female youth. These findings are consistent with existing literature that highlights the clustering of HIV infections in areas with limited access to healthcare services, high population densities, and socio-economic disparities [68]. To validate these spatial patterns, Moran’s I statistic and Geary’s C statistic confirmed the presence of spatial autocorrelation, demonstrating that HIV prevalence is not randomly distributed but rather clustered in specific locations. Addressing these spatial inequalities requires targeted interventions in high-risk areas, such as rural and peri-urban settings, where female youth may face barriers to accessing sexual and reproductive health services. Our findings suggest that spatial variations in HIV prevalence are influenced not only by individual-level determinants but also by broader neighbourhood-level factors. The inclusion of spatial random effects in our model accounts for unmeasured contextual influences, such as community-level healthcare access, socio-economic disparities, and local HIV prevention efforts. This highlights the importance of considering both individual behaviours and structural determinants when designing targeted interventions.

The spatial analysis identified two HIV hotspots in the study area, reinforcing the need for geographically focused public health efforts. Cluster 1, located near Greater Edendale, showed an HIV prevalence of 48.4%, a relative risk (RR) of 1.22, and a *p*-value of 0.025, indicating a 22% higher risk inside the cluster compared to outside. Cluster 2, covering Nadi, KwaMbanjwa, and the surrounding areas, had an HIV prevalence of 49.6% and an RR of 1.28 but was not statistically significant (*p* = 0.467).

The findings of this study are consistent with previous research on HIV determinants in South Africa. Similar studies have found that educational attainment, economic status, and healthcare access play crucial roles in shaping HIV risk. However, the incorporation of spatial modelling techniques in this study provides a unique perspective on geographic variations in HIV prevalence, adding depth to the existing literature.

Globally, studies in sub-Saharan Africa have also reported spatial clustering of HIV infections, particularly in regions with poverty, gender inequality, and limited health infrastructure. This highlights the importance of context-specific interventions tailored to regional disparities.

These findings highlight the need for targeted public health interventions, particularly for young women in high-prevalence areas. Strengthening community-based prevention programmes can help address both behavioural and structural risk factors, while expanding HIV testing and counselling services will improve early diagnosis and linkage to care. Increasing access to Pre-Exposure Prophylaxis (PrEP) and antiretroviral therapy (ART) in underserved regions is essential for reducing new infections and improving health outcomes. Additionally, socio-economic empowerment initiatives, including education and employment programmes, should be prioritised to enhance resilience against HIV. Policymakers should focus on targeted resource allocation and integrating HIV prevention with economic support programmes to mitigate structural inequalities. Strengthening school-based HIV education can further promote safer sexual practices, while enhanced spatial surveillance of HIV trends will optimise intervention planning and improve public health strategies.

## 5. Contributions of This Study

This study advances existing research by applying Bayesian spatial modelling to analyse HIV prevalence among young females in KwaZulu-Natal. Unlike traditional regression models, this approach accounts for spatial autocorrelation, offering deeper insights into geographic patterns of HIV risk. Additionally, while many studies focus solely on individual-level factors, this research integrates spatial, socio-demographic, and economic determinants, providing a more comprehensive understanding of HIV risk. Compared to studies in developed nations, where HIV prevalence is lower, and healthcare access is widespread, this study highlights unique challenges in a high-burden, resource-limited setting, reinforcing the need for context-specific interventions.

## 6. Implications of the Study Findings

These findings have significant public health implications for HIV prevention in KwaZulu-Natal. The spatial clustering of HIV prevalence underscores the need for targeted interventions in high-risk rural and peri-urban areas with limited healthcare access. Socio-economic determinants, such as education and income, highlight the potential impact of economic empowerment programmes and improved educational access in reducing HIV risk among young women. Furthermore, the association between risky sexual behaviours and HIV prevalence reinforces the need for behavioural interventions, including comprehensive sexuality education, condom distribution, and expanded PrEP access for vulnerable populations.

## 7. Strengths and Limitations

This study has several strengths. The use of a Bayesian spatial logistic regression model provided robust estimates of HIV risk while accounting for spatial dependencies and heterogeneity. The high-resolution geographic data from the HIV Incidence Provincial Surveillance System (HIPSS) enhanced the ability to identify high-risk areas with precision. Additionally, the inclusion of socio-demographic, behavioural, and economic determinants allowed for a comprehensive understanding of HIV risk factors among females aged 15–34 in KwaZulu-Natal. Rigorous quality control mechanisms, including real-time data monitoring and automated anomaly detection, strengthened the reliability of the dataset.

However, some limitations must be acknowledged. Although HIPSS employed a robust sampling strategy, potential selection biases or underreporting could still affect data representativeness. Additionally, missing data were addressed using complete-case analysis, which led to the exclusion of 19.8% of cases. While this approach ensured analytical consistency, it may have introduced bias if excluded individuals differed systematically from those retained in the analysis. Furthermore, self-reported behavioural data (e.g., condom use and alcohol consumption) may have been affected by social desirability bias, potentially influencing the accuracy of reported behaviours and estimates of risk factors. Stigma surrounding HIV testing and disclosure may have also contributed to underreporting of HIV prevalence in certain areas, particularly in rural communities. Additionally, the study’s spatial resolution, while detailed, may not fully capture urban–rural variations in HIV dynamics, necessitating more granular spatial analyses in future research.

## 8. Future Research Directions

Future studies should explore alternative methods for handling missing data, such as multiple imputation, to assess its impact on HIV prevalence estimates. Future studies should build on these findings by incorporating longitudinal data to establish causal relationships between socio-demographic factors, spatial effects, and HIV prevalence rather than merely identifying associations. Additionally, longitudinal analyses can help evaluate the long-term effectiveness of targeted interventions and track changes in HIV prevalence over time. Expanding this research to other regions in South Africa or sub-Saharan Africa would improve the generalisability of findings and help identify common spatial patterns of HIV risk. Additionally, integrating qualitative methods, such as community-based interviews or focus groups, could provide deeper insights into the social and cultural factors shaping HIV risk, complementing quantitative analyses. Future research should also focus on stratified spatial analyses for the 15–19 age group to identify emerging transmission patterns and potential “hot spots” for new infections. Examining trends in this younger population across multiple survey rounds would help determine whether the observed lower prevalence reflects a true decline in new infections or a delay in risk exposure. By tailoring interventions to the specific needs of female youth in high-risk locations, policymakers can enhance HIV prevention efforts and mitigate the epidemic’s impact in KwaZulu-Natal.

## 9. Conclusions

This study highlights the significant spatial variation in HIV prevalence among female youth in KwaZulu-Natal, South Africa, and identifies key demographic, behavioural, and health-related risk factors. The Bayesian spatial logistic regression approach enabled the integration of spatial effects and covariates, providing a nuanced understanding of HIV risk in this population. Key findings indicate that young adults aged 20–34, particularly those with lower educational attainment and limited economic opportunities, face higher odds of HIV infection. Risky behaviours, including alcohol use, multiple sexual partnerships, and inconsistent condom use, further increase vulnerability. Additionally, co-infections with TB and STIs significantly elevate the risk of HIV. Conversely, tertiary education, salaried employment, consistent condom use, and legal marriage serve as protective factors, underscoring the need for multi-sectoral approaches to HIV prevention. Given the observed spatial heterogeneity, geographically targeted interventions are essential to address localised drivers of HIV risk. Strategies should prioritise improving education, promoting economic empowerment, expanding access to HIV prevention services, and integrating co-infection management within healthcare programmes. This study underscores the importance of addressing both individual-level and geographic factors in HIV prevention efforts. By tailoring interventions to identified spatial clusters, public health programmes can enhance their effectiveness and reduce the burden of HIV in KwaZulu-Natal.

## Figures and Tables

**Figure 1 ijerph-22-00446-f001:**
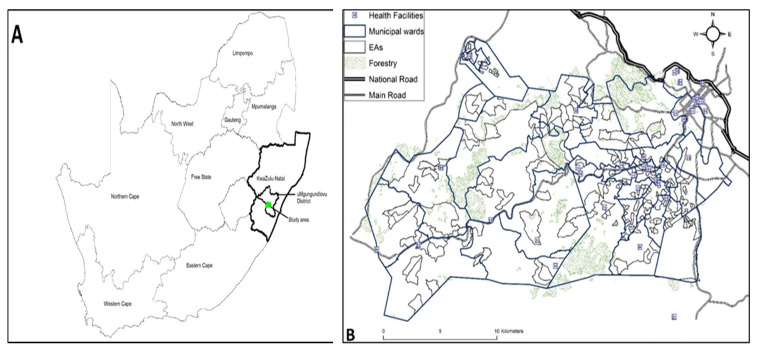
(**A**,**B**) location of the study area.

**Figure 2 ijerph-22-00446-f002:**
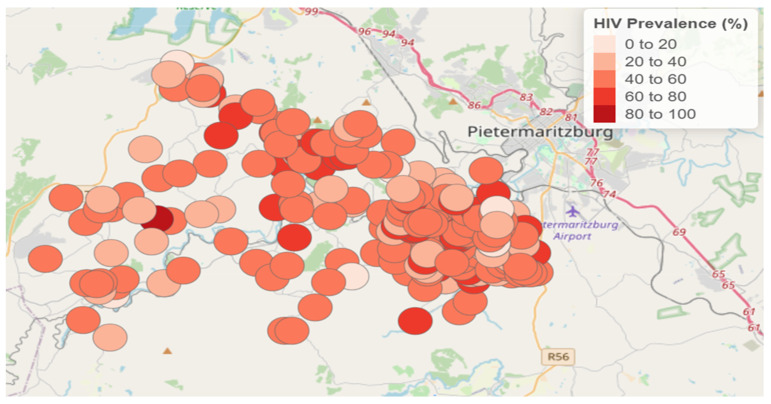
Geographical distribution of unsmoothed HIV prevalence among enumeration areas.

**Figure 3 ijerph-22-00446-f003:**
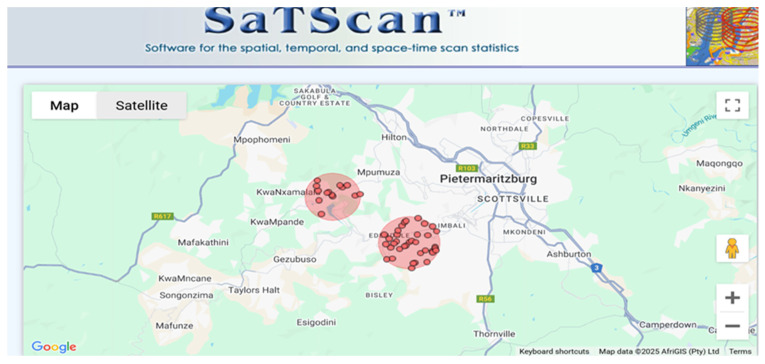
Spatial clustering of HIV prevalence in uMgungundlovu Municipality.

**Figure 4 ijerph-22-00446-f004:**
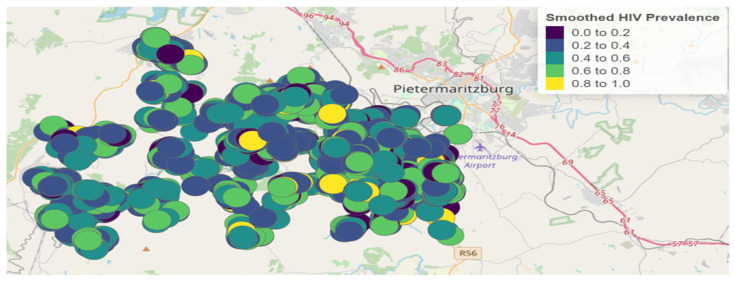
Geographical distribution of smoothed HIV prevalence rates.

**Figure 5 ijerph-22-00446-f005:**
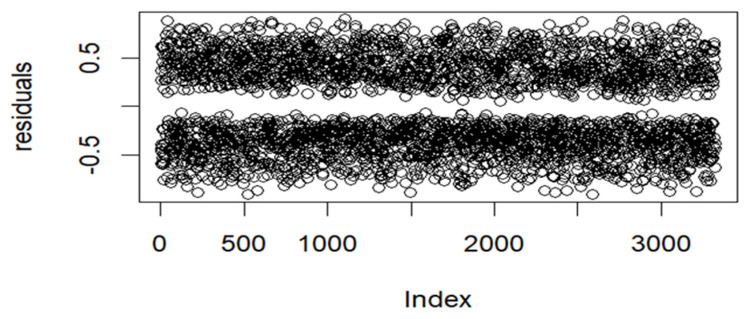
Residuals plot for the fitted model.

**Figure 6 ijerph-22-00446-f006:**
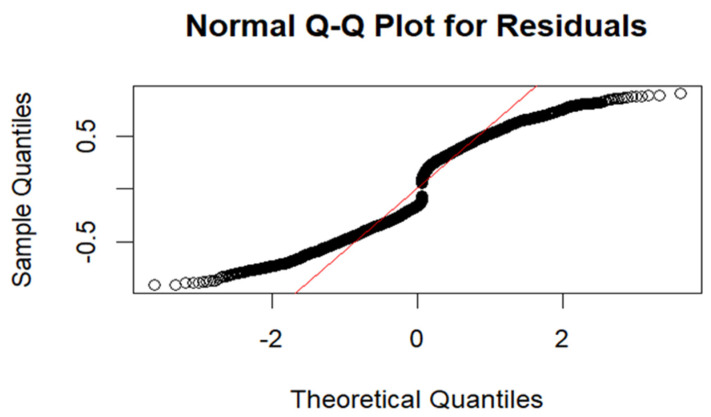
Normal Q–Q plot for the residuals.

**Figure 7 ijerph-22-00446-f007:**
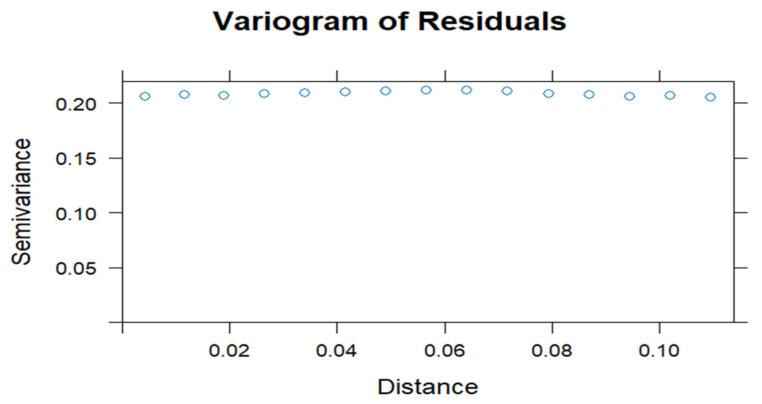
Variogram plot for the residuals.

**Figure 8 ijerph-22-00446-f008:**
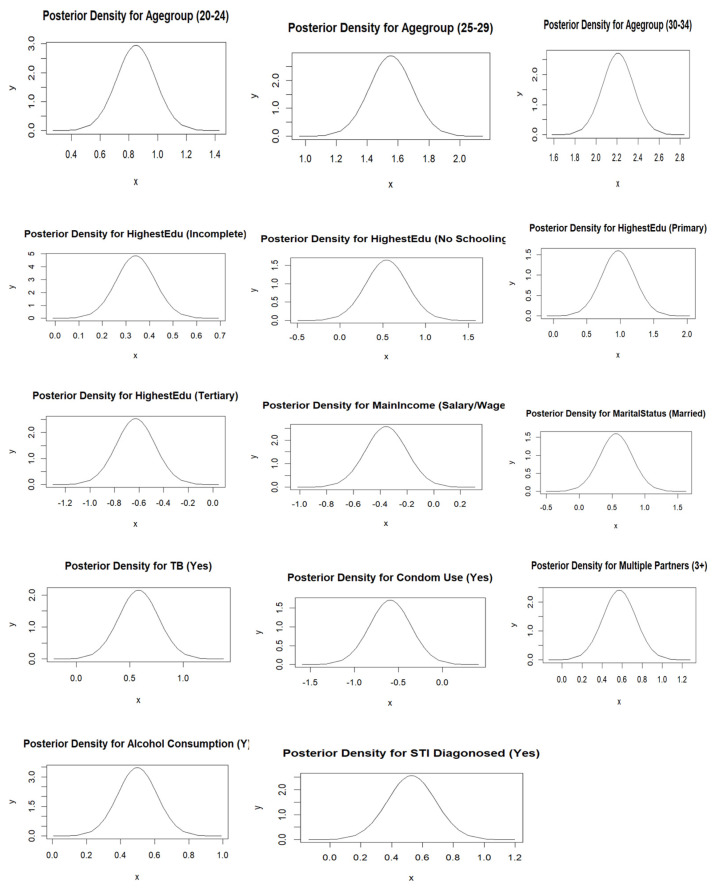
Posterior density plots for statistically significant coefficients in the model.

**Table 1 ijerph-22-00446-t001:** Unweighted HIV prevalence rates by covariate among HIV-positive female youth in Vulindlela and Greater Edendale areas in uMgungundlovu Municipality.

Covariate	n = 1576	HIV Prevalence (%)	95% CI Lower	95% CI Upper	p-Value
**Age Group**					
15–19	88	20.4	16.8	24.5	<0.0001
20–24	399	37.0	34.2	40.0	
25–29	546	54.0	50.8	57.1	
30–34	543	67.5	64.2	70.8	
**Ever Pregnant**					
No	282	37.4	33.9	40.9	<0.0001
Yes	1294	50.4	48.4	52.3	
**Education Level**					
Complete secondary	737	44.3	41.9	46.7	<0.0001
Incomplete secondary (Grades 8–11/NTC1/2)	660	52.1	49.3	54.9	
No response	0	0.00	0.00	97.5	
No schooling/creche/pre-primary	45	55.6	44.1	66.6	
Primary (Grades 1–7)	60	70.6	59.7	80.0	
Tertiary (diploma/degree)	74	32.9	26.8	39.4	
**Main Income**					
No income	102	50.5	43.4	57.6	0.169432
No response	36	49.3	37.4	61.3	
Other	0	0.00	0.00	97.5	
Other non-farming income	102	47.9	41.0	54.8	
Pension or grants	541	50.4	47.4	53.5	
Remittance (migrant worker sending money home)	40	50.0	38.6	61.4	
Salary and/or wage	748	44.8	42.4	47.3	
Sales of farming products	7	50.0	23.0	77.0	
**Marital Status**					
Divorced	2	100.0	15.8	100.0	0.000181
Legally married	70	38.0	31.0	45.5	
Living together like husband and wife	56	51.4	41.6	61.1	
Separated but still legally married	2	100.0	15.8	100.0	
Single and never been married/never Lived together as husband/wife before	1357	47.0	45.2	48.8	
Single but have been living with someone as husband/wife before	86	63.7	55.5	71.8	
Widowed	3	60.0	14.7	94.7	
**Ever diagnosed with TB**					
No	1482	46.7	45.5	48.5	0.000365
No response	2	28.6	36.7	71.0	
Yes	92	63.0	54.6	70.8	
**Condom use**					
No	50	58.1	47.0	68.7	0.056253
Yes	1526	47.1	45.4	48.9	
**Number of sexual partners**					
1	1278	45.5	43.6	47.3	<0.0001
2	159	51.6	45.9	57.3	
3+	139	67.5	60.6	73.8	
**Alcohol consumption**					
No	1326	45.8	43.9	47.6	<0.0001
Yes	250	58.5	53.7	63.3	
**Ever diagnosed with STI**					
No	1438	46.3	44.5	48.1	<0.0001
Yes	138	63.0	56.2	69.4	
**Forced first sex**					
Do not remember	26	54.2	39.2	68.6	0.246837
No	1503	47.1	45.4	48.9	
Yes	47	54.7	43.5	65.4	
**Away from home**					
No	1391	47.0	45.2	48.9	0.407053
No response	7	58.3	27.7	84.8	
Yes	178	50.1	44.8	55.5	
**Length in community**					
Always	1196	46.7	44.8	48.7	0.447987
Moved here less than 1 year ago	62	48.1	39.2	57.0	
Moved here more than 1 year ago	315	50.1	46.1	54.1	
No response	3	60.0	14.7	94.7	
**Accessed health care**					
Did not respond	2	33.3	4.3	77.7	0.018296
No	950	45.6	43.4	47.8	
Yes	624	50.5	47.7	53.4	
**Run out of money**					
Did not respond	34	45.9	34.3	57.9	0.618173
No	1206	47.0	45.1	49.0	
Yes	336	49.1	45.2	52.9	
**Meal cuts**					
Did not respond	28	40.6	28.9	53.1	0.515632
No	1259	47.5	45.6	49.4	
Yes	289	47.7	43.7	51.8	

**Table 2 ijerph-22-00446-t002:** Moran’s I and Geary’s C summary statistics.

Summary Statistics	Moran’s Index	Geary’s C
Statistic	0.707	0.291
*p*-value	<2.2 × 10^−16^	<2.2 × 10^−16^
Expectation	−0.0003052	1.000000
Variance	0.0001070	0.0001434
Standard Deviate	68.361	59.176

**Table 3 ijerph-22-00446-t003:** Model selection criteria summary for the two competing models.

Spatial Logistic Model	DIC	pD	$\tilde{D}$	WAIC
Unstructured	4128.952	48.89294	4080.059	4129.874
Structured	4127.739	40.20267	4087.537	4128.783

**Table 4 ijerph-22-00446-t004:** Adjusted Odds Ratios and 95% credible intervals for the parameters of the structured model.

Covariate	OR	95% CI Lower	95% CI Upper
Intercept	0.289	0.055	1.517
**Age Group (ref: 15–19)**			
20–24	2.337	1.791	3.053
25–29	4.745	3.611	6.234
30–34	9.198	2.883	12.293
**Education (ref: Complete Secondary)**			
Incomplete secondary (Grades 8–11/NTC1/2)	1.405	1.195	1.652
No response	0.800	0.129	4.968
No schooling/creche/pre-primary	1.718	1.065	2.773
Primary (Grades 1–7)	2.612	1.597	4.276
Tertiary (diploma/degree)	0.534	0.391	0.728
**Main Income (ref: No Income)**			
No response	0.827	0.473	1.445
Other	0.793	0.129	4.899
Other non-farming income	0.862	0.575	1.294
Pension or grants	0.813	0.595	1.111
Remittance	0.987	0.576	1.689
Salary and/or wage	0.706	0.522	0.956
Sales of farming products	0.815	0.301	2.203
**Marital Status (ref: Divorced)**			
Living together like husband and wife	0.731	0.289	1.850
Legally married	0.371	0.150	0.919
Single and never been married/never lived together as husband before	0.959	0,399	2.307
Separated but still legally married	1.781	0.328	9.650
Single but have been living with someone as husband before	1.354	0.539	3.401
Widowed	0.840	0.200	3.518
**Ever pregnant (ref: No)**			
Yes	1.137	0.939	1.374
**Run out of money (ref: Did not respond)**			
No	0.965	0.578	1.611
Yes	0.977	0.566	1.687
**Meal cuts (ref: Did not respond)**			
No	1.398	0.825	2.370
Yes	1.197	0.681	2.106
**TB (ref: Never Suffered from TB)**			
No response	0.665	0.182	2.430
Yes	1.799	1.247	2.594
**Condom Use (ref: No)**			
Yes	0.552	0.348	0.874
**Number of Sexual Partners (ref: 1)**			
2	1.212	0.936	1.568
3+	1.765	1.275	2.445
**Alcohol (ref: No)**			
Yes	1.644	1.310	2.063
**STI Diagnosed (ref: No)**			
Yes	1.694	1.245	2.303
**Forced First Sex (ref: Do not remember)**			
No	0.767	0.433	1.357
Yes	1.070	0.528	2.168
**Away From Home (ref: No)**			
No response	1.328	0.476	3.706
Yes	1.244	0,975	1.586
**Length in Community (ref: Always)**			
Moved here less than 1 year ago	1.011	0.689	1.486
Moved here more than 1 year ago	0.983	0.806	1.201
No response	1.699	0.438	6.586
**Accessed Health Care (ref: Did not respond)**			
No	1.292	0.440	3.789
Yes	1.576	0.536	4.637

## Data Availability

The dataset used in this research is available upon reasonable request, from the corresponding author. However, restrictions apply to these data’s availability and are not publicly available due to maintaining participants’ confidentiality.

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
