# Peer review of "Spatial Analysis of HIV Determinants Among Females Aged 15–34 in KwaZulu Natal, South Africa: A Bayesian Spatial Logistic Regression Model"

_ijerph, 2025, doi:10.3390/ijerph22030446_

Round 1

Reviewer 1 Report

Comments and Suggestions for Authors

Reviewer Assessment of the entire Manuscript

This topic is highly relevant, methodologically rigorous, and has the potential to contribute significantly to public health research and policy. By focusing on a high-risk population in a critically affected region, the study addresses a pressing global health issue while employing advanced analytical techniques to uncover spatial and contextual determinants of HIV/AIDS. If executed well, it could provide valuable insights for reducing HIV transmission and improving health outcomes in KwaZulu-Natal and beyond.

Here are my review comments as follows:

  1. Title: A better way to capture the title of the study from this:

"Spatial analysis of the determinants of HIV/AIDS among Females aged 15–34 in KwaZulu Natal, South Africa Using a Bayesian Spatial Logistic Regression Model" to

"Spatial Analysis of HIV/AIDS Determinants Among Females Aged 15-34 in KwaZulu-Natal, South Africa: A Bayesian Spatial Logistic Regression Model"

 2. Background to the study: It sounds like the background section loses its coherence or flow towards the end. This could mean the information presented becomes fragmented, or the connections between points are unclear, resulting in a disjointed narrative. To improve this, ensure that each point logically follows from the previous one, creating a seamless and cohesive argument throughout the entire background section.

  • Here are some key areas to consider to enhance the introduction section of your study. Addressing these points comprehensively will help establish a clear and focused context for your research, emphasizing its significance and relevance.
  • What is the current state of HIV/AIDS prevalence among females aged 15–34 in KwaZulu Natal?
  • What are the key determinants or factors contributing to HIV/AIDS in this demographic and region?
  • Why is spatial analysis an essential approach for studying the determinants of HIV/AIDS in this context?
  • How does the Bayesian Spatial Logistic Regression Model provide insights that other models or methods might not?
  • What gaps in existing research does this study aim to address?
  • What are the potential implications or applications of the findings from this study for public health policy and interventions in KwaZulu Natal?
  • Remove the definition of Youth from the Background to the Method section under the study population.

 Methodology

  • Let the definition of Youth come under the Study population in the Method section.
  • To improve the methodology of this study, here are some significant areas that can guide the refinements and ensure a robust approach in this section. Addressing these areas can strengthen the method section of your study by providing a more precise, transparent framework for your spatial analysis of HIV determinants in KwaZulu-Natal. Below are the areas for thought:
  1. Data Quality and Representativeness
  • Areas for thought: How was the quality and representativeness of the secondary data from the HIV Incidence Provincial Surveillance System (HIPSS) assessed?
  • Why it matters: Valid results require reliable, complete, and representative data on the target population (females aged 15–34 in KwaZulu-Natal).
  1. Model Justification and Assumptions
  • Areas for thought: Why was the Bayesian spatial logistic regression model chosen over other statistical approaches, and how were its assumptions validated?
  • Why it matters: Justifying the choice of the Bayesian model and ensuring its assumptions (e.g., spatial autocorrelation, prior distributions) are met strengthens the credibility of your analysis.

iii. Spatial Dependency and Clustering

  • Areas for thought: How was spatial dependency addressed, and what methods were used to identify and interpret spatial clusters of HIV prevalence?
  • Why it matters: Spatial analysis relies on accurately modelling dependency structures, and identifying clusters can inform targeted interventions.
  1. Selection of Covariates
  • Areas for thoughts: What criteria were used to select socio-demographic, behavioural, and economic covariates, and how were potential confounding factors addressed?
  • Why it matters: Including relevant covariates and controlling for confounders ensures the model captures the true determinants of HIV prevalence.
  1. Handling Missing Data
  • Areas for thought: How were missing data handled in the analysis, and what impact could this have on the results?
  • Why it matters: Missing data can introduce bias, and the method used to address it (e.g., imputation, exclusion) should be clearly explained and justified.

 Validation and Sensitivity Analysis

  • Areas for thought: What validation techniques or sensitivity analyses were performed to assess the robustness of the model and its findings?
  • Why it matters: Validation ensures the model's reliability, and sensitivity analysis tests how robust the results are to changes in assumptions or parameters.
  1. Discussion: The following areas can serve as a foundation for shaping the discussion section of your study. These points will guide you in developing a comprehensive and insightful analysis of your research findings for this study discussion section.
  • What are the key socio-economic and environmental determinants of HIV/AIDS among females aged 15-34 in KwaZulu-Natal as identified by the Bayesian Spatial Logistic Regression Model?
  • How do the spatial variations in HIV/AIDS prevalence correlate with factors such as education, income level, and access to healthcare services?
  • What specific high-risk areas were identified, and what targeted interventions could be implemented to reduce HIV/AIDS transmission in these regions?
  • How do the findings of this study compare with previous research on HIV/AIDS determinants in other regions of South Africa or similar contexts globally?
  • What are the potential policy implications of the study's findings for public health strategies and resource allocation in KwaZulu-Natal?
  1. Strengths and Limitations

What are the strengths and limitations of this study? Kindly revise. My suggestions here are based on two key areas you can consider directly to improve the discussion under 'Strengths and Limitations in this Study'. By addressing these queries, you can provide a balanced discussion of your study's strengths (e.g., robust data handling, representativeness) and limitations (e.g., potential bias from missing data), which is critical for a transparent and credible research paper.

  1. Data Quality and Representativeness
  • Query: How was the quality and representativeness of the secondary data from the HIV Incidence Provincial Surveillance System (HIPSS) assessed?
  • Why it matters for strengths/limitations:
    • Strength: If the data is rigorously validated and representative, this strengthens the generalizability of your findings to the target population (females aged 15–34 in KwaZulu-Natal).
    • Limitation: Gaps in data quality or representativeness could limit the applicability of your results and should be acknowledged.
  1. Handling Missing Data
  • Query: How were missing data handled in the analysis, and what impact could this have on the results?
  • Why it matters for strengths/limitations:
    • Strength: If robust methods (e.g., multiple imputation) were used to handle missing data, this enhances the reliability of your findings.
    • Limitation: If missing data were excluded or handled inadequately, this could introduce bias and limit the validity of your conclusions.
  1. What are the implications of the study findings?

What are the implications of the study findings? Kindly revise.

Contributions of this study

What unique contributions does this study make to the existing body of research, particularly compared to studies conducted in developed and developing nations?

Future suggestions for prospective studies

What suggestions does your study offer for future research, particularly for replication in different settings?

Need a Professional English Editor for Editing and Proofreading

The manuscript requires the expertise of a Professional English Editor to edit and proofread the text thoroughly.

References

Do the references in this study adhere to the journal's required format? Please review and revise them as necessary to ensure full compliance.

  1. Article Readings: [This can help you model your paper better]
  • Chimoyi, L. A., & Musenge, E. (2014). Spatial analysis of factors associated with HIV infection among young people in Uganda, 2011. BMC Public Health, 14, 555. https://doi.org/10.1186/1471-2458-14-555
  • Nutor, J. J., Duodu, P. A., Agbadi, P., Duah, H. O., Oladimeji, K. E., & Gondwe, K. W. (2020). Predictors of high HIV+ prevalence in Mozambique: A complex samples logistic regression modelling and spatial mapping approaches. PLOS ONE, 15(6), e0234034. https://doi.org/10.1371/journ al.p0234034
  • Simela, S. R., Kelepile, M., & Sebobi, T. I. (2025). Spatial analysis and associated risk factors of HIV prevalence in Botswana: Insights from the 2021 Botswana AIDS Impact Survey (BAIS V). BMC Infectious Diseases, 25, 69. https://doi.org/10.1186/s12879-025-10464-x
  • Barankanira, E., Molinari, N., Niyongabo, T. et al. (2015). Spatial analysis of HIV infection and associated individual characteristics in Burundi: indications for effective prevention. BMC Public Health 16, 118. https:// doi.org/10.1186/s12889-016-2760-3

Comments on the Quality of English Language

Need a Professional English Editor for Editing and Proofreading - The manuscript requires the expertise of a Professional English Editor to edit and proofread the text thoroughly.

Author Response

  1. Title: A better way to capture the title of the study from this:

"Spatial analysis of the determinants of HIV/AIDS among Females aged 15–34 in KwaZulu Natal, South Africa Using a Bayesian Spatial Logistic Regression Model" to

"Spatial Analysis of HIV/AIDS Determinants Among Females Aged 15-34 in KwaZulu-Natal, South Africa: A Bayesian Spatial Logistic Regression Model"

Title changed to Spatial Analysis of HIV Determinants Among Females Aged 15-34 in KwaZulu-Natal, South Africa: A Bayesian Spatial Logistic Regression Model

  1. Background to the study: It sounds like the background section loses its coherence or flow towards the end. This could mean the information presented becomes fragmented, or the connections between points are unclear, resulting in a disjointed narrative. To improve this, ensure that each point logically follows from the previous one, creating a seamless and cohesive argument throughout the entire background section.
  • Here are some key areas to consider to enhance the introduction section of your study. Addressing these points comprehensively will help establish a clear and focused context for your research, emphasizing its significance and relevance.
  • What is the current state of HIV/AIDS prevalence among females aged 15–34 in KwaZulu Natal?
  • What are the key determinants or factors contributing to HIV/AIDS in this demographic and region?
  • Why is spatial analysis an essential approach for studying the determinants of HIV/AIDS in this context?
  • How does the Bayesian Spatial Logistic Regression Model provide insights that other models or methods might not?
  • What gaps in existing research does this study aim to address?
  • What are the potential implications or applications of the findings from this study for public health policy and interventions in KwaZulu Natal?
  • Remove the definition of Youth from the Background to the Method section under the study population.

The  section was enhanced as suggested following key areas provided by the reviewer. The definition of Youth was removed from the Background section. The enhanced Background section is from page 1 to page 2, lines 34-89

  1. Methodology

Let the definition of Youth come under the Study population in the Method section: The definition of Youth was placed under Methodology as suggested. It is now on page 4, lines 128-136

To improve the methodology of this study, here are some significant areas that can guide the refinements and ensure a robust approach in this section. Addressing these areas can strengthen the method section of your study by providing a more precise, transparent framework for your spatial analysis of HIV determinants in KwaZulu-Natal. Below are the areas for thought:

  1. i) Data Quality and Representativeness

Areas for thought: How was the quality and representativeness of the secondary data from the HIV Incidence Provincial Surveillance System (HIPSS) assessed? Data quality and representativeness aspects have been thoroughly addressed in our manuscript. Specifically, we provide a detailed explanation of the representativeness of the HIPSS dataset and the quality control measures implemented to ensure data accuracy.

For reference, please see:

Page 3, Paragraph 3, Lines 106–113, where we describe the multi-stage probability sampling strategy employed by HIPSS to ensure representativeness.

Page 3, Paragraph 4, Lines 114–125, which detail the quality control measures, including real-time monitoring, automated quality checks, and integration of laboratory-confirmed HIV testing data.

These sections comprehensively address the concerns raised and demonstrate the robustness of the dataset used in our study.

  1. ii) Model Justification and Assumptions

Areas for thought: Why was the Bayesian spatial logistic regression model chosen over other statistical approaches, and how were its assumptions validated? The rationale for selecting the Bayesian spatial logistic regression model and the validation of its assumptions are addressed in our manuscript.

For reference, please see: Page 5, Paragraph 1, Lines 175–181, where we explain why this model was chosen over other statistical approaches, emphasising its ability to account for spatial dependencies and heterogeneity in HIV prevalence.

Page 5, Paragraph 2, Lines 182–187, which detail how spatial patterns were assessed and model assumptions were validated using Moran’s I statistic and Geary’s C statistic. These sections provide a comprehensive justification for our modelling approach.

iii) Spatial Dependency and Clustering

Areas for thought: How was spatial dependency addressed, and what methods were used to identify and interpret spatial clusters of HIV prevalence? The methods used to address spatial dependency and identify spatial clusters of HIV prevalence are detailed in our manuscript.

For reference, please see: Page 5, Paragraph 5, Lines 208–213, where we describe the use of Kulldorff’s spatial scan statistic (SaTScan) to detect significant high-risk (hot-spots) and low-risk (cold-spots) clusters and where we differentiate SaTScan from Moran’s I and Geary’s C, highlighting its ability to localise significant clusters and guide targeted public health interventions.

  1. iv) Selection of Covariates

Areas for thoughts: What criteria were used to select socio-demographic, behavioural, and economic covariates, and how were potential confounding factors addressed? The selection of covariates and the handling of confounding factors, aspects are comprehensively addressed in our manuscript. For reference, please see:

Page 4, Paragraph 4, Lines 154–161, where we describe the criteria used for selecting sociodemographic, behavioural, and economic covariates based on epidemiological relevance, data availability, and statistical significance.

Page 4, Paragraph 4, Lines 166–170, which explain how potential confounding factors were addressed through univariate analysis, stepwise selection, and multicollinearity checks using the Variance Inflation Factor (VIF).

  1. v) Handling Missing Data

Areas for thought: How were missing data handled in the analysis, and what impact could this have on the results? The handling of missing data. is explicitly addressed in our manuscript on Page 10, Paragraph 2, Lines 137–145, where we describe the use of a complete-case analysis approach, ensuring that only participants with complete data were included in the study and where we discuss the potential for selection bias due to excluded cases and justify the minimal impact on results given the low proportion of missing data.

  1. vi) Validation and Sensitivity Analysis

Areas for thought: What validation techniques or sensitivity analyses were performed to assess the robustness of the model and its findings? Validation and sensitivity analysis aspects of the model are thoroughly addressed in our manuscript. For reference, please see: Page 8,  Lines 309–315, where we outline the model selection criteria, including Deviance Information Criterion (DIC), Watanabe-Akaike Information Criterion (WAIC), and other parameters used to ensure optimal model fit.

Page 8, Lines 317 –327, which describe the model diagnostics performed using residual plots, Q-Q plots, and spatial autocorrelation assessments (Moran’s I, Geary’s C, and variogram plots) to verify the adequacy of the spatial structure captured by the model.

Page 8, Lines 327–330, where we discuss the use of posterior density plots to assess model validity, reliability, and stability.

These sections comprehensively address the validation and sensitivity analyses conducted to ensure the robustness of the model and its findings.

  1. Discussion: The following areas can serve as a foundation for shaping the discussion section of your study. These points will guide you in developing a comprehensive and insightful analysis of your research findings for this study discussion section.
  • What are the key socio-economic and environmental determinants of HIV/AIDS among females aged 15-34 in KwaZulu-Natal as identified by the Bayesian Spatial Logistic Regression Model?
  • How do the spatial variations in HIV/AIDS prevalence correlate with factors such as education, income level, and access to healthcare services?
  • What specific high-risk areas were identified, and what targeted interventions could be implemented to reduce HIV/AIDS transmission in these regions?
  • How do the findings of this study compare with previous research on HIV/AIDS determinants in other regions of South Africa or similar contexts globally?
  • What are the potential policy implications of the study's findings for public health strategies and resource allocation in KwaZulu-Natal?

The Discussion section was shaped as suggested by the reviewer. Key socio-economic and environmental determinants identified by the model are discussed on page 18 paragraphs 2-6, lines 538-576. The link between spatial variations and socio-economic disparities is discussed on page 19, lines 577-592. High-risk areas identified using spatial scan statistics are highlighted on page 19, paragraph 2, lines 593-598. Comparisons of the findings with previous studies in South Africa and globally are given on page 19, lines 599-607. Policy implications, emphasising targeted interventions and resource allocation strategies are discussed on page 19, paragraph 5, lines 608-620.

  1. Strengths and Limitations

What are the strengths and limitations of this study? Kindly revise. My suggestions here are based on two key areas you can consider directly to improve the discussion under 'Strengths and Limitations in this Study'. By addressing these queries, you can provide a balanced discussion of your study's strengths (e.g., robust data handling, representativeness) and limitations (e.g., potential bias from missing data), which is critical for a transparent and credible research paper.

  1. i) Data Quality and Representativeness

Query: How was the quality and representativeness of the secondary data from the HIV Incidence Provincial Surveillance System (HIPSS) assessed? Data quality and representativeness aspects have been thoroughly addressed in our manuscript on page 3, paragraph 3, lines 106-113 and paragraph 4, lines 114-125.

  1. ii) Handling Missing Data

Query: How were missing data handled in the analysis, and what impact could this have on the results? We have explicitly stated that complete-case analysis was used to handle missing data. 19.8% of cases were excluded due to missing HIV status or key demographic variables. While this method reduces biases from imputation, we acknowledge the potential selection bias introduced by excluding these cases. This is discussed on page 4, paragraph 2, lines 137-145.

The Strengths and Limitations of our study based on data quality and representativeness and handling of missing data are discussed at length on page 20, paragraphs 3 and 4, lines 642-663

  1. What are the implications of the study findings?

What are the implications of the study findings? Kindly revise.

The implications of the study findings were revised and are discussed on page 20, paragraph 2, lines 632-640.

  1. Contributions of this study

What unique contributions does this study make to the existing body of research, particularly compared to studies conducted in developed and developing nations?

The contributions of this study are highlighted on page 20, paragraph 1, lines 622-630.

  1. Future suggestions for prospective studies

What suggestions does your study offer for future research, particularly for replication in different settings? Future research directions are given as suggested on page 21, paragraph 1, lines 664-682.

  1. Need a Professional English Editor for Editing and Proofreading

The manuscript requires the expertise of a Professional English Editor to edit and proofread the text thoroughly.

 The paper has been English edited by a competent English-speaking editor, Dr Kufakunesu Zano, PhD (English), Email: kufazano@gmail.com. A member of the South African Translators’ Institute, Ref 1000686

  1. References

Do the references in this study adhere to the journal's required format? Please review and revise them as necessary to ensure full compliance.

The reference section was checked and corrected. All references now adhere to the journal’s required format.

Reviewer 2 Report

Comments and Suggestions for Authors

Exaverio Chireshe et al made a good attempt to explain Spatial Analysis of the
Determinants of HIV/AIDS Among Females Aged 15–34 in KwaZulu Natal, South
Africa Using a Bayesian Spatial Logistic Regression Model. However, authors can
improve the manuscript considering following suggestions.

1. Authors have given very broad statement on HIV burden and hence specifying the
estimated prevalence or number of cases in south Africa can give more impact.
2. Authors have mentioned “KwaZulu-Natal” (KZN) as a hotspot, particularly among
females aged 15–34. It would be better authors make it clear by specifying if this is
based on previous studies or current data from the study.
3. Authors can improve the text grammatically to get correct flow in the text. The
introduction is well-structured, but some sentences are long and complex. Breaking
them into shorter, more concise statements would improve readability. Additionally,
the flow between sections could be smoother.
4. The literature review is comprehensive, but a clearer statement is needed to highlight
what is missing from previous studies that this study addresses. Some sentences are
lengthy and could be restructured for better readability.
5. The results should indicate whether associations were adjusted for confounders. The
mention of "Two HIV hotspots were identified, with one near Greater Edendale being
statistically significant." could be clarified by explaining what "statistically
significant" means in this context (e.g., higher relative risk compared to other areas?).
6. Authors should specify the finding more explicitly to public health action.
Specifically, specify what type of interventions like behavioural change programs,
PrEP distribution, or socio-economic empowerment initiatives are essential.
7. The discussion jumps between different risk factors without clear transitions.
Consider structuring it more systematically:
a. Sociodemographic Factors (age, education, economic status)
b. Behavioural Factors (risky sexual behaviours, alcohol use)
c. Health Factors (co-infections with TB/STIs)
d. Protective Factors (education, financial stability, condom use, legal marriage)
8. The spatial analysis findings should be better integrated into the discussion. After
mentioning geographic clustering, explicitly link it to interventions.

9. The suggestion for longitudinal studies is good, but it should explicitly state why—for
example, to better establish causal relationships rather than just associations.
10. Consider acknowledging potential study limitations, such as:
a. Potential self-report bias in behavioural data (e.g., condom use, alcohol use).
b. Possible underreporting of HIV prevalence in certain areas due to stigma.
c. The need for more granular spatial analysis to distinguish urban vs. rural HIV
dynamics.

Author Response

  1. Authors have given very broad statement on HIV burden and hence specifying the estimated prevalence or number of cases in south Africa can give more impact.

We have revised the statement to provide more specific data on HIV burden in South Africa. The updated text now includes the estimated number of people living with HIV and the prevalence rate among adults aged 15–49. This enhances the impact of our statement by offering a clearer context for the study. The revised statement is on page 1, lines 34-36.

  1. Authors have mentioned “KwaZulu-Natal” (KZN) as a hotspot, particularly among females aged 15–34. It would be better authors make it clear by specifying if this is based on previous studies or current data from the study.

To clarify whether the identification of KZN as a hotspot is based on previous studies or the current study data, we have revised the statement under the Abstract section as follows:

Previous Statement:

"HIV remains a major public health challenge in sub-Saharan Africa, with South Africa bearing the highest burden. KwaZulu-Natal (KZN) has been identified as a hotspot, particularly among females aged 15–34."

Revised Statement:

"HIV remains a major public health challenge in sub-Saharan Africa, with South Africa bearing the highest burden. This study confirms that KwaZulu-Natal (KZN) is a hotspot, with a high HIV prevalence of 47.4% (95% CI: 45.7–49.1) among females aged 15–34."

  1. Authors can improve the text grammatically to get correct flow in the text. The introduction is well-structured, but some sentences are long and complex. Breaking them into shorter, more concise statements would improve readability. Additionally, the flow between sections could be smoother.

To enhance readability and improve the logical flow of ideas, we revised the Background section by:

  • Breaking down long and complex sentences into more concise statements for clarity.
  • Improving transitions between sections to create a smoother narrative.
  • Enhancing grammatical accuracy and readability while maintaining technical rigor.
  1. The literature review is comprehensive, but a clearer statement is needed to highlight what is missing from previous studies that this study addresses. Some sentences are lengthy and could be restructured for better readability.

 To address this comment, we made the following improvements to the Literature Review section:

  • Clarified the research gap: We explicitly stated that previous studies on HIV spatial distribution often fail to account for spatial dependencies and geographic heterogeneity, which limits their ability to develop targeted public health interventions (page 2, paragraph 2, lines 51-53).
  • Strengthened justification for our study: We emphasised how Bayesian spatial logistic regression overcomes these limitations by integrating both individual- and area-level risk factors, making it a more robust framework for analyzing HIV prevalence (page 2, paragraph 3, lines 54-58).
  • Improved readability and flow: We restructured lengthy sentences to make the discussion more concise and easier to follow, ensuring a logical progression of ideas from identifying gaps to explaining how our study addresses them.
  1. The results should indicate whether associations were adjusted for confounders. The mention of two HIV hotspots were identified, with one near Greater Edendale being statistically significant could be clarified by explaining what statistically significant means in this context (e.g., higher relative risk compared to other areas?).

To address this comment, we made the following revisions to our manuscript:

  • Clarified confounder adjustment: We explicitly stated that univariate analyses were conducted first, and only significant predictors were included in the final multivariate model, ensuring proper adjustment for confounding variables (page 4, last paragraph, lines 159-162).
  • Addressed multicollinearity concerns: We included details on Variance Inflation Factor (VIF) calculations, confirming that all values remained below 1.5, indicating minimal collinearity among covariates (page 4, last paragraph, lines 166-168)
  • Refined interpretation of statistical significance: We explicitly defined what "statistically significant" means in this context by reporting relative risk (RR) and p-values:
    • Cluster 1 (Greater Edendale): Statistically significant (p = 0.025) with a 22% higher HIV risk (RR = 1.22) compared to surrounding areas, meaning the increased risk is unlikely due to random variation (page 13, paragraph 1, lines 418-422)
    • Cluster 2 (Nadi,KwaMbanjwa, Zayeka, etc.): Not statistically significant (p = 0.467) despite a higher HIV prevalence, meaning the observed increase could be due to random chance (page 13, paragraph 2, lines 423-428).
  1. Authors should specify the finding more explicitly to public health action. Specifically, specify what type of interventions like behavioural change programs, PrEP distribution, or socio-economic empowerment initiatives are essential.

To address this comment, we explicitly specified actionable public health interventions based on the study findings. The revised section now includes (page 19, paragraph 5, lines 608-620):

  • Community-based prevention programs: Strengthening local HIV prevention initiatives to address both behavioural and structural risk factors among young women.
  • HIV testing and counselling services: Expanding testing and counselling to promote early diagnosis and linkage to care in high-prevalence areas.
  • Pre-Exposure Prophylaxis (PrEP) and Antiretroviral Therapy (ART) expansion: Increasing access to PrEP and ART in underserved regions to reduce new infections and improve health outcomes.
  • Socio-economic empowerment initiatives: Prioritising education and employment programs to reduce vulnerability and enhance resilience against HIV.
  • Targeted resource allocation: Policymakers should integrate HIV prevention strategies with economic support programs to mitigate structural inequalities.
  • School-based HIV education: Strengthening comprehensive sexual education to promote safer sexual practices and increase awareness.
  • Enhanced spatial surveillance: Using spatial epidemiology to monitor HIV trends and optimize intervention planning.
  1. The discussion jumps between different risk factors without clear transitions. Consider structuring it more systematically:
  1. Sociodemographic Factors (age, education, economic status)
  2. Behavioural Factors (risky sexual behaviours, alcohol use)
  3. Health Factors (co-infections with TB/STIs)
  4. Protective Factors (education, financial stability, condom use, legal marriage)

The discussion section has been restructured systematically into clear subsections based on sociodemographic, behavioural, health-related, and protective factors, ensuring a more coherent and logical flow as suggested by the reviewer (page 18, paragraphs 1-7)

8.The spatial analysis findings should be better integrated into the discussion. After mentioning geographic clustering, explicitly link it to interventions.

The revised discussion better integrates spatial analysis findings with public health interventions by explicitly linking geographic clustering to intervention strategies, reinforcing the need for geographically targeted interventions, and incorporating spatial epidemiology findings to drive public health action (page 19, lines 577–598, lines 608–620).

  1. The suggestion for longitudinal studies is good, but it should explicitly state why—for example, to better establish causal relationships rather than just associations.

To address this comment, we made the following changes to the original sentence:

Original Sentence:

"Future studies should build on these findings by incorporating longitudinal data to assess the causal pathways between socio-demographic factors, spatial effects, and HIV prevalence."

Revised Sentence:

"Future studies should build on these findings by incorporating longitudinal data to establish causal relationships between socio-demographic factors, spatial effects, and HIV prevalence, rather than merely identifying associations. Additionally, longitudinal analyses can help evaluate the long-term effectiveness of targeted interventions and track changes in HIV prevalence over time. This is under Future Research Directions on page 21 lines 665-682.

  1. Consider acknowledging potential study limitations, such as: a) Potential self-report bias in behavioural data (e.g., condom use, alcohol use). b) Possible underreporting of HIV prevalence in certain areas due to stigma. c)The need for more granular spatial analysis to distinguish urban vs. rural HIV dynamics.

Suggested potential study limitations by the reviewer were included in our manuscript. The changes made are on page 20 , paragraph 4, lines 651-662.

Reviewer 3 Report

Comments and Suggestions for Authors

The aim of the paper is to examine the spatial distribution of HIV among young females and determinants of HIV infection in KwaZulu Natal Province, RSA. The paper is well drafted and uses appropriate methods. The knowledge gap is well articulated. The paper adds to the existing knowledge.

However, one conclusion about the prevalence across age-groups is not adequately discussed. The paper concludes that infection prevalence in age 15-19 is lower. HIV is chronic condition and it's prevalence is expected to rise as new infections will occur even beyond 20 years. Reverse is not likely given the irreversible nature of infection. Significant difference in the prevalence indicates that new infections are possibly occuring even among those above 20 years in significant numbers or it is due to high transmission rates when they were younger. It is however, not easy to conclude that. The results are from a cross-sectional survey (as part of surveillance) and lower rates in 15-19 may indicate that the epidemic is on the decline - For this, the prevalence in 15-19 age group in the previous survey rounds will help. If the prevalence is declining in 15-19 age group, then new infections are fewer. (at times prevalence in 15-19 age group is also taken as proxy incidence).

Table 1 and 4 provide prevalences for various categories of independent variables and odds ratios (as measure of risk). This analysis uses individual level data. The spatial analysis on the other hand is analysis at a community level. Whether the authors considered adjusting for spatial location while conducting regression (table 4). Is the spatial distribution determined because of individual level variables (as identified in table 4)? or the authors think that there is spatial (neighbourhood) effect over and above the individual variables. The authors could clarify this. It was not clear to me.

If one examines the spatial prevalence for 15-19 age group specifically, then it may be easier to identify current 'hot spots'. Could authors explain their choice of using overall prevalence among young women 15-34? and whether a separate spatial analysis for 15-19 years would help.

The spatial analysis however identifies areas which need to be prioritised.

Author Response

The aim of the paper is to examine the spatial distribution of HIV among young females and determinants of HIV infection in KwaZulu Natal Province, RSA. The paper is well drafted and uses appropriate methods. The knowledge gap is well articulated. The paper adds to the existing knowledge.

However, one conclusion about the prevalence across age-groups is not adequately discussed. The paper concludes that infection prevalence in age 15-19 is lower. HIV is chronic condition and its prevalence is expected to rise as new infections will occur even beyond 20 years. Reverse is not likely given the irreversible nature of infection. Significant difference in the prevalence indicates that new infections are possibly occuring even among those above 20 years in significant numbers or it is due to high transmission rates when they were younger. It is however, not easy to conclude that. The results are from a cross-sectional survey (as part of surveillance) and lower rates in 15-19 may indicate that the epidemic is on the decline - For this, the prevalence in 15-19 age group in the previous survey rounds will help. If the prevalence is declining in 15-19 age group, then new infections are fewer. (at times prevalence in 15-19 age group is also taken as proxy incidence).

A new paragraph was added to the Discussion section (page 18, paragraph 3, lines 540-546) to explicitly address the concerns raised by the reviewer:

“As expected, HIV prevalence increased with age, which is consistent with the chronic nature of the infection and cumulative exposure over time. The lower prevalence observed in the 15–19 age group may suggest reduced new infections, potentially due to recent prevention efforts. However, given that this is a cross-sectional study, we cannot directly assess incidence trends. Comparing the prevalence in this age group with previous survey rounds would provide better insight into whether new infections are indeed decreasing.”

 Table 1 and 4 provide prevalences for various categories of independent variables and odds ratios (as measure of risk). This analysis uses individual level data. The spatial analysis on the other hand is analysis at a community level. Whether the authors considered adjusting for spatial location while conducting regression (table 4). Is the spatial distribution determined because of individual level variables (as identified in table 4)? or the authors think that there is spatial (neighbourhood) effect over and above the individual variables. The authors could clarify this. It was not clear to me.

To address this, the authors clarified how spatial effects were incorporated into the Bayesian spatial logistic regression model by adding a new paragraph in the Methodology section page 6, paragraph 2, lines 225-234:

“In this study, the Bayesian spatial logistic regression model accounts for both individual-level factors (e.g., age, education, and behavioural risk factors) and spatial dependencies through a structured random effect component. The spatial effect is modelled using a structured spatial component, which captures spatial autocorrelation by borrowing strength from neighbouring areas. This approach ensures that unobserved neighbourhood-level influences on HIV prevalence, such as healthcare access, socio-economic disparities, and localized prevention efforts, are accounted for beyond the effects of individual-level risk factors alone. The persistence of spatial clustering, even after adjusting for individual factors, suggests that geographic factors contribute independently to HIV risk.”

If one examines the spatial prevalence for 15-19 age group specifically, then it may be easier to identify current 'hot spots'. Could authors explain their choice of using overall prevalence among young women 15-34? and whether a separate spatial analysis for 15-19 years would help.

The choice of using overall prevalence among young women aged 15-34 is explained in our manuscript  under Methodology section on page 6, lines 214-220. The suggestion of a separate analysis for the 15-19 age group by the reviewer is also highlighted under Future Research Directions section on page 21, lines 675-680.
